# What is Normalization? The Strategies Employed in Top-Down and Bottom-Up Proteome Analysis Workflows

**DOI:** 10.3390/proteomes7030029

**Published:** 2019-08-22

**Authors:** Matthew B. O’Rourke, Stephanie E. L. Town, Penelope V. Dalla, Fiona Bicknell, Naomi Koh Belic, Jake P. Violi, Joel R. Steele, Matthew P. Padula

**Affiliations:** 1Bowel Cancer & Biomarker Lab, Northern Clinical School, Faculty of Medicine and Health, The University of Sydney Lvl 8, Kolling Institute. Royal North Shore Hospital, St. Leonards, NSW 2065, Australia; 2School of Life Sciences and Proteomics Core Facility, Faculty of Science, The University of Technology Sydney, Ultimo 2007, Australia; 3Respiratory Cellular and Molecular Biology, Woolcock Institute of Medical Research, The University of Sydney, Glebe 2037, Australia

**Keywords:** normalization, top down proteomics, 2D-PAGE, LC-MS/MS

## Abstract

The accurate quantification of changes in the abundance of proteins is one of the main applications of proteomics. The maintenance of accuracy can be affected by bias and error that can occur at many points in the experimental process, and normalization strategies are crucial to attempt to overcome this bias and return the sample to its regular biological condition, or normal state. Much work has been published on performing normalization on data post-acquisition with many algorithms and statistical processes available. However, there are many other sources of bias that can occur during experimental design and sample handling that are currently unaddressed. This article aims to cast light on the potential sources of bias and where normalization could be applied to return the sample to its normal state. Throughout we suggest solutions where possible but, in some cases, solutions are not available. Thus, we see this article as a starting point for discussion of the definition of and the issues surrounding the concept of normalization as it applies to the proteomic analysis of biological samples. Specifically, we discuss a wide range of different normalization techniques that can occur at each stage of the sample preparation and analysis process.

## 1. Introduction

Proteomics is a field that encompasses a multitude of techniques and technologies that are applied to a variety of scientific questions. However, it is quite often the case that only a subset of the techniques is applied to answer a specific question because the techniques come with limitations, such as the inability to use harsh surfactants when studying protein complexes and interactions which could result in insolubility and an inability to analyze all complexes. One of the most common applications of proteomics is for the quantification of the abundance of ‘proteins’, a term that encompasses proteoforms, Open Reading Frame (ORF) products and proteins complexes. Due to the sheer number of distinct proteoforms, their range of abundances and the variability of this abundance in different cells or tissues or environmental conditions, the accurate quantification of abundance changes is challenging and subject to bias and errors.

Normalization is defined as the process of returning something to a normal condition or state. In system-wide -omics analysis, normalization tries to account for bias or errors, systematic or otherwise, to make samples more comparable [1] and measure differences in the abundance of molecules that are due to biological changes rather than bias or errors. When referring to normalization strategies used in proteomics, whether it be gel based or Liquid Chromatography-Mass Spectrometry (LC-MS)-based, a great deal of work has been performed to develop software solutions that attempt normalization towards the end of an acquisition, using either gel densitometry images or ion intensity values from technical replicates. At this point, normalization is to overcome differences in staining intensity or ionization efficiency that are often beyond the direct control of the researcher. However, many points in the experimental process occur before this in which normalization of some type could be applied to make the samples more comparable. Normalization is traditionally understood to be a function that is performed post acquisition of data to account for random variance and “batch effects” (which will be discussed later). However, when considering the actual function of normalization, enabling proper proportionate comparison of different biological samples, the methods that can achieve this are highly varied. Simple approaches such as a performing protein level quantitation prior to sample digestion, can easily be considered as normalization steps. Additionally, robustness can also be introduced through normalization of samples by increasing the reproducibility of the measurements within technical replicates. A further consideration is that normalization needs to be applied across biological replicates and treatments. With all these considerations in mind, the term normalization in proteomics and indeed other ‘omics’ style system wide analyses, becomes more of a “strategy” or experimental design approach than a single technique. Therefore, when one is deciding how to go about ensuring the best possible removal of bias and systematic error, the appropriate methodologies available are numerous but which should be applied is unclear.

In our own work, we have often wondered whether the strategies we apply throughout our experimental work to minimize variation and normalize data are actually achieving that objective. The wide range of biological samples that pass through our Core Facility means that some of the strategies developed by other researchers to normalize data in specific samples or situations do not apply to the samples that we are analyzing. In this article, we review the means of normalization and the potential multi technique approaches that can be used to achieve a normalized workflow while also pointing out misgivings we have with the way those means and workflows are applied. In some cases, we have no solutions but would like to flag the issue to the greater proteomics community so that some form of consensus may be reached.

## 2. Cellular Normalization

The quantification of proteins by relative abundance or fold-change in comparison to a reference or control is arguably no longer sufficient to infer cellular events of biological significance [2,3]. Therefore, absolute quantification based on protein copies per cell has become a desirable objective in recent years [4]. The key advantage of absolute quantification is the ability to compare absolute values across multiple published data sets, though this is currently unachievable. In addition, the normalization of this is a technical challenge due to many confounding factors, such as the estimation of cell number in a sample, the degree of cell variability and the changing protein expression between individual cells within the same population. Although it may be argued that these problems can be overcome with methods such as Stable Isotope Labelling by Amino acids in Cell culture (SILAC) [5], where label incorporation occurs in the cell and therefore allows cellular level normalization, the use of this technique incurs an additional expense per experiment and it is susceptible to many of the issues outlined below such as what parameter to normalize to. Additionally, label-free quantitative methods are also desirable due to the greater range of proteome coverage achieved when more than one peptide is required for identification [6].

As highlighted by Wiśniewski et al. (2014) [3], current methods of determining protein copy number per cell require an accurate estimation of cells in the sample. In microbial culturing, a common method of measuring cells is via the Optical Density (OD) at 600 nm, whereby the amount of light scattered is proportional to the number of cells present. However, it has been shown that a significant bias is introduced in the form of fluorescent reporter proteins that absorb light at the same wavelength [7]. While it has been suggested that this can be normalized by measuring OD at the alternate wavelength of 700 nm, the remaining issue with this method is that it constitutes an estimate of cell abundance rather than an absolute value [8]. Furthermore, cell viability is not taken into account, which is particularly problematic for experiments in which treatments or conditions may significantly impact microbial viability [9].

In tissue culture procedures, manual counting utilizing a haemocytometer remains a method commonly employed by many laboratories due to its cost-effective nature and wide applicability to virtually any homogenous cell culture. However, due to the arduous nature of this task, estimates are based on a relatively low sampling size and are subjective to the laboratory analyst. Automated image-based cell counters reduce this bias considerably but are reliant on software algorithms that must accurately identify a cell type based on cell size and morphology. Therefore, aggregates of cells, which can be characteristic of many cell lines, are unlikely to be detected by these algorithms [10]. Alternatively, flow cytometry with commercially available counting beads allows rapid and accurate quantification of cells, with the additional advantage of being able to differentiate between cell types in complex or heterogeneous samples [11]. However, this technique is less versatile due to reduced availability of relevant bead standards, which are considerably expensive in comparison to manual techniques. In addition, this technique requires a high level of technical expertise. Another caveat is that a single-cell suspension is fundamental for flow cytometry, and therefore, significant processing of tissues and adherent cell lines is required to avoid the formation of aggregates [12].

There are numerous methods to either count or estimate the number of cells in a sample. However, none of these methods typically account for the inherent variabilities present within a homogeneous cell population. For example, estimations based on cell volume fail to take into account differences in cell size, which in turn is affected by cell cycle stage [13]. These issues can be circumvented by the “proteomic ruler” approach. This methodology shows that there is a proportional relationship between DNA mass, cell counts, and the cumulative MS signal of histone-derived peptides. Therefore, accurate measurement of DNA content within a proteomic sample can allow this relationship to be exploited to predict the copy numbers per cell of an individual protein based on its MS signal [3]. A rapid method for measurement of DNA is Absorbance at 260 nm (A260), although the accuracy of this measurement can be compromised by the presence of protein, meaning that DNA must be effectively isolated from the sample. However, the purity of the DNA can be assessed by calculating the A260/A280 ratio, whereby a ratio of approximately 1.8 is indicative of an uncontaminated DNA sample [14]. Therefore, while the proteomic ruler is a promising normalization method it does not account for the difference in protein abundance between cells of the same population which result from stochastic or pulsatile gene expression [15]. In addition, it is well-documented that histone mRNA and protein biosynthesis are increased 35-fold during the S-phase of the cell cycle [16]. As such, the ratio of cells in S-phase within a culture at the time of harvest could result in an artificially higher histone MS signal and calculated cell count. Furthermore, this disparity is increased in complex samples, such as whole tissue lysates with multiple cell types, or mixed samples whereby there is more than one organism present. Importantly, whether these are legitimate issues with the proteomic ruler or not has not been determined. A practice that can be used to mitigate potential S-phase fluctuations skewing data is to work with homogenous cell lines that are synchronized by serum starvation prior to treatments, keeping the supplemental fetal calf serum at a low percentage [17], inducing cell senescence or if feasible working with terminally differentiated cells. However, this is not useful for studying cell lines normally utilized for experiments or animal models.

Similar normalization methods based on genomic DNA copy numbers have also emerged, particularly in relation to metabolomics [18]. These, like the proteomic ruler, rely on the assumption that the genome size and ploidy is known. This may pose an issue with plant matter and some microorganisms due to polyploidy or conversion between ploidy states [19,20], exemplifying a bias of these normalization protocols towards mammalian systems. In these circumstances, flow cytometry is required to determine ploidy. However, flow cytometry does not account for conversion between ploidy states, which is known to be affected by several factors [21]. In addition, copy number variation, which is widely reported in disease states [22], would significantly influence such normalization methods.

As mentioned, SILAC is a highly reliable method for protein quantification and has been reported to have minimal errors [23]. The errors that can occur are the in vivo conversion of labelled arginine into to other amino acids (typically proline [24]), isotopic amino acids not incorporating completely, and errors with sample mixing. The conversion of arginine to proline can negatively influence the accuracy of protein quantification. The extent of labelling and the amino acid conversion profile could be problematic for primary cells as many of these do not proliferate well and consequently may not achieve consistent label incorporation for particular proteins, which could lead to further errors. Another solution that has been presented is to incorporate label-swap replications to increase the reliability of expression ratios, but this would need to be used synergistically with post analysis statistical normalization which has its own limitations [25].

## 3. Normalization During and after Lysis and Protein Extraction

A key part of any proteomics workflow is sample disruption for the extraction of protein. How these steps are conducted depends heavily on the experimental aim, the separation technology to be used (electrophoresis or chromatography, for instance), and the chemistry of the sample [26,27]. It is outside the scope of this review to describe all conceivable strategies. However, it is significant to note that the chemicals used and handling of these initial stages of sample disruption are well-documented to affect downstream results significantly (carbamylation of lysine by excessive heating of urea solutions for example [28]), and therefore the uniformity and type of sample handling at these stages is important when considering the effectiveness of normalization strategies [29].

There are several critical points in cell lysis: Protein extraction, separation, solubilization and removal of unwanted contaminants or molecules, and each is a point where bias can be introduced. The workflow is as follows, using cell lines as an example: the sample is harvested from the growth media and centrifuged. It can then be washed and resuspended in the relevant lysis buffer [30] before progressing to protein extraction and ultimately to analysis. The first step must be completed at a consistent time point amongst biological replicates and samples (typically as measured by cellular normalization strategies) to avoid carrying over variance in growth time that could affect normalization strategies during and after lysis and protein extraction [31]. The reason is due to biological variances, such as different growth times, usually carrying a greater bias effect than technical variance [32].

Furthermore, an important consideration is to limit the presence or occurrence of random chemical modifications that can be introduced during sample handling. Endogenous proteases can often be a source of random cleavage events which will in turn effect the efficacy of subsequent deliberate proteolytic cleavage [33]. The approach most commonly used in proteomics is to add protease inhibitors to samples as soon as tissues are dissociated [34] or in the case of liquid based samples, at the point of collection [35].

In our own work, we assume that once the lysis and protein extraction protocol chosen has been deemed fit for purpose, uniform and accurate handling is the most reliable method of limiting technical variance at this stage. Nonetheless, more research into this area regarding other normalization strategies at the time of or just after sample disruption would be beneficial, as there is currently little to no experimental studies explicitly evaluating how normalization is affected by these factors at this early stage.

## 4. Pre-Digestion/Digestion Normalization

### 4.1. Normalization Using Total Protein Quantification

The accurate quantification of the total amount of protein is essential in all proteomics experiments. In Isoelectric Focusing-based Two-Dimensional Polyacrylamide Gel Electrophoresis (IEF-based 2D-PAGE), all replicate gels need to be loaded with equivalent amounts of total protein, and this also applies to Top Down MS workflows where samples undergo multidimensional separations prior to LC-MS/MS [36]. In peptide-centric approaches, total protein is used to calculate the ratio of enzyme to protein required. The normalization of the total protein amount ensures the same amount of protein is being digested by the same amount of proteolytic enzyme under controlled conditions in order to achieve consistency in protein cleavage. There are multiple methods and commercially available kits for the quantification of proteins that have been widely used and reviewed [37,38,39]. These techniques are fluorometric, colorimetric or densitometry-based assays that make assumptions on the samples being quantified, as compared in Table 1. One point that should be raised is that these methods are typically very easy to contaminate and can provide highly unreliable and non-reproducible concentrations.

The assay of choice to ascertain total protein amount needs to be chosen with careful consideration. Many of these can be unreliable in the presence of reducing agents or surfactants frequently used in the preparation of samples prior to protein digestion, while the assay needs to be sensitive enough to not consume all of the available sample [37]. The primary colorimetric assays include the Bicinchoninic Acid Assay (BCA) [40], the Bradford or Coomassie blue assay [41], and the Lowry assay [42], while the main fluorescent-based assays are the Qubit protein assay or those measuring tryptophan fluorescence [43]. A number of these methods quantify the presence of one or a select group of amino acids in a protein or peptide. This generates bias in the quantification of samples as no two proteins have the same ratio of amino acids. The test also assumes that differences will be averaged out amongst the overall population of proteins. The BCA method, for instance, relies on the reduction of Cu^2+^ in the reagent to Cu^+^ in the presence of the amino acids cysteine, tyrosine and tryptophan, the quantity of which will vary in different proteoforms. A secondary reaction then quantifies these amino acids by producing a purple color in the presence of Cu^+^ [40]. Likewise, the measurement of A280 uses the absorption properties of the aromatic rings in tyrosine, tryptophan and phenylalanine as a quantitative measurement to quantitate all the protein in the sample [44]. Both of these assays are sensitive to the presence of surfactants. One method that has been both traditionally successful and more recently shown to be highly sensitive is the use of Coomassie brilliant blue staining coupled with 1D SDS PAGE [45]. This approach uses a known protein concentration to generate a standard curve “in Gel” that is stained with Coomassie brilliant blue stain and then scanned and measured with either an optical of fluorescence scanner. The advantage to using such a system, despite the inconvenience, is that it is resistant to many of the interferences that other methods would otherwise be susceptible to, such as surfactants, as these molecules are washed out of the gel.

### 4.2. Digestion Optimization to Reduce Bias

Trypsin commonly does not offer complete digestion of proteins and it is well-documented that trypsin often miscleaves in sequences of multiple lysine and arginine amino acids [46]. When analyzed, the peptide fragments that are detected in the MS could originate from peptides with missed cleavages thereby increasing the complexity of the bottom-up data, the potential for false positive identifications, and bias in quantification. This will be further discussed later in the review.

For enzymatic digests, the optimal conditions for the enzymatic reaction need to be taken into account. Trypsin, for instance, works optimally at a pH of 8.0 and a temperature of 37 °C. The enzyme to protein ratio has a range of 1:20 to 1:100 depending on the length of reaction time. Methods to increase both the efficiency and the reaction time are varied and even include microwaving the samples [47]. The inclusion of additional enzymes, such as Lys C in combination with trypsin is another such way of increasing the efficiency of the cleavage of the proteins into peptides [48]. It also needs to be acknowledged that the analysis of samples using a single protease enzyme generates a protease bias. It has been demonstrated that when samples are analyzed by different proteolytic enzymes the identified proteins, sequence coverage and quantification are altered [49,50]. Nevertheless, whichever quantification method is chosen to capture the “completeness” of digestion per sample, the act of optimizing digestion to achieve an equalized level of proteolysis is critical when normalizing sample preparation.

### 4.3. Monitoring Digestion Efficiency

Digestion efficiency monitoring is an overlooked step for normalization that could be accounting for some biologically-attributed variability, especially in samples that have become resistant to proteases by the action of protein expression changes or by Post-Translational Modifications (PTMs) such as dityrosine crosslinking [51] or modification of lysine by methylation or acetylation. Testing batch variation for the efficiency of the digestion enzyme is not often performed and is left to the manufacturer to guarantee the enzyme quality. Variability within batches and across large sample cohorts can be tested using an external test of the enzyme by fluorescence colorimetric substrate assay kits [52]. Additionally, enzyme efficiency could also be affected by multiple cycles of freezing and thawing beyond the manufacturers’ recommendation.

One method has been proposed to accurately gauge whether the digestion efficiency in a given sample has worked is to place a minute amount of each sample into a tube for co-reaction with a fluorescently labelled substrate. The fluorescent readings are then taken over the course of digestion [53].

### 4.4. Normalization Issues Unique to Top Down Methodologies

The normalization of spot density across the multiple gels necessary for 2D-PAGE is a well-documented process, and many points outlined for normalizing sample amounts in general also apply to 2D-PAGE. There is a crucial necessity to ensure that the total amount of protein loaded on each Immobilized pH Gradient strip (IPG strip) is identical so that the density of the same spot across technical replicate gels is also identical. After image alignment and determination of which is the same spot across different gels, normalization is performed using the density of a spot across different technical replicates. This process is then repeated for all spots. The density of a spot across different treatments is then used to determine whether a spot is a proteoform that differs in its abundance compared to the control. In an attempt to reduce bias due to technical variation, Difference Gel Electrophoresis (DiGE) was developed to allow samples to be loaded and separated on the same gel. However, DiGE does not change the need to ensure that an equal amount of each sample is mixed prior to IEF and may experience differences in labelling efficiency in different samples. Preferential labelling has been reported [54,55,56] and labelling could be altered if PTMs block the chemical group on a proteoform that the dye molecule reacts with or if pH is not in the correct range [57]. Protein reference standards are available for 2D-PAGE but their use is rare.

For a number of years, it was assumed that each spot contained only one (or at least very few) proteoforms. This was because subsequent analysis of peptides produced by in-gel digestion of a spot in the MS tended to produce one hit with numerous peptides, and maybe a few extra additional hits with a single peptide identified. Recently, the use of instrumentation with a much higher sensitivity has provided support against this assumption [58,59], suggesting that spot intensity is due to numerous unrelated proteoforms altering in abundance in different ways.

## 5. Post-digestion Normalization

### 5.1. Peptide Assays for as a Means of Post-Digestion Normalization

Though protein assays are versatile tools which can be performed at differing points throughout a workflow containing a complicated series of chemical reactions, the majority of articles report assaying total protein amount just after sample extraction and just prior to further sample handling [60,61,62,63]. Thus, there is an assumption that either no losses are occurring or losses affect all samples equally and cancel out. This could be a dangerous assumption if labelling technologies such as Isobaric Tags for Relative and Absolute Quantification (iTRAQ) or Tandem Mass Tags (TMT) are being used, because relative quantification relies on equal amounts of labelled peptides being mixed [64,65]. SILAC could be equally affected regardless of whether samples are mixed at the protein or peptide level. For peace of mind and to ensure samples are still equalized and normalized, peptide assays can be utilized as a means of post-digestion normalization.

A peptide assay detects the total concentration of peptides in solution in a complex sample, just as a protein assay detects the total concentration of proteins [43]. There are a range of assays available which have been optimized for peptides, such as the Nanodrop relative quantification platforms from ThermoFisher, as well as CBQ peptide-level assays [66]. Otherwise, colorimetric and fluorometric assays exist which are essentially modified protein assays that are optimized for the detection of peptides rather than proteins. To their advantage, these peptide-assays are typically 3-to 4-fold more sensitive than comparable protein assays, use a smaller working volume and detect smaller amounts of peptides in the complex mixture, including concentrations as low as 25 µg/mL. One issue with colorimetric assays is that the same amount of sample is being loaded but not necessarily the same amount of peptides due to incomplete protein digestion, and the assay can react with undigested proteins as well as peptides produced by proteolysis. For example, the BCA assay indicates the presence of amino acids irrespective of whether the amino acids are in a peptide or a protein. This could be circumvented through the utilization of a Molecular Weight Cut-Off (MWCO) filter, as employed in Filter Aided Sample Preparation (FASP) workflows [67], to remove non-digested proteins, and then quantify the sample peptide concentrations. However, this does result in unrecoverable peptide loss because of incomplete digestion or the potential for peptides to irreversibly bind to the filter and bias the subsequent analysis.

Normalization via a peptide assay acts the same way as normalization via a protein assay. By understanding the concentration of peptides per sample at the end of sample preparation for the MS, the loading amount for the MS can then be adjusted so that the amount of peptide per sample is as equivalent as possible. In this way, a peptide assay acts as an important means of normalization to ensure that the same amount of peptide is loaded into the analysis instrument and potentially increase the accuracy of downstream quantification as it can account for bias introduced through sample handling after assaying the total amount of protein.

Intriguingly, the use of peptide-assays in place of or in addition to protein assays in bottom-up proteomic workflows is not common, making an assessment of their use as a normalization tool difficult. A review of the literature regarding bottom-up proteomics workflows found that a protein assay was the more commonly applied tool [68,69]. If a peptide assay was applied, more often, a protein assay was also applied. The small uptake is likely due to time and cost reasons, as well as the assumption that the relative benefit for using a peptide assay, even for a new sample, is low. In our own experience of performing both protein and peptide assays, we consistently found good agreement between the two assays and no overall improvement in our quantification efficiency, suggesting that the benefit from adding this test is not sufficient to warrant the additional time and cost for every experiment. However, assaying peptide amount prior to LC-MS/MS has revealed samples that have less peptide than expected from conducting a protein assay earlier in the experimental process. The use of both a protein assay and a peptide assay may be worthwhile to note any biological particularities relevant in an unfamiliar sample when conducting initial tests and removed when no longer providing a benefit to save time and cost [70]. Insights into the biology of the sample, such as the amount of protein digestion with a particular enzyme, or the effect of a particular clean-up protocol, can also be assessed using this method. Thus, again, the aim of the experiment, the sample type, and fitness-for-purpose of the assays are the three important considerations driving decision-making in this regard [29].

### 5.2. Normalization using Internal Standards

Post-digestion normalization is also achievable using spike-ins and endogenous molecules. Spiking samples with internal standards (ISTDs) is common practice in all LC-MS based fields of science. Normalization using an ISTD/s spiked during post-digestion sample preparation is important to account for differences between samples which may have occurred due to instrumentation faults such as retention time drifts and sensitivity fluctuations. These ISTDs or ‘spike-ins’ are peptide analogues with the same or very similar physicochemical properties to the peptides in the sample that would not be naturally occurring, such as heavy-labelled peptide analogues. The addition of individual or multiple heavy-labelled peptides at a known concentration post-digestion can allow for absolute quantification while also allowing inter-run normalization. This is because the concentration and abundance of this spike-in is likely to be uniform across different samples and therefore can act as a baseline for any instrument fluctuations that may have occurred. However, this does not account for bias introduced before the addition of the ISTD. While individual ISTDs can be used for specific peptides of interest, a mix of ISTDs is typically applied when looking at an entire proteomic sample. One example of a commercially available ISTD mix is Biogynosys’s Indexed Real Time (iRT) peptides kit, containing eleven non-naturally occurring synthetic peptides [71]. Multiple software packages including Spectronaut™ normalize the data acquired in these runs based on the slight retention time shifts and intensity drifts of these eleven iRT peptides.

The addition of heavy-labelled proteins of known concentration to a protein extract is potentially one of the most reliable forms of normalization. This method allows absolute quantification of specific peptides and thus a protein (ORF product in peptide-centric workflows) in a sample regardless of ionization efficiencies. It also allows normalization across injections in the MS. In support of this, an article by Russell & Lilley (2012) [72] found that using heavy-labelled purified proteins was the most effective form of normalization, in comparison to label-free and iTRAQ methods. The addition of labelled proteins at the start of a workflow, instead of adding heavy-labelled peptides at the end as is commonly practiced, was found to produce the smallest source of technical variance because all proteins were then subjected to identical handling up to and including analysis in the MS [72]. However, in peptide-centric workflows, the normalization is specific to ORF products rather than proteoforms unless the peptide defining a proteoform is detected. The addition of a heavy-labelled peptide or protein representing an analogue for every proteoform present in a sample is not only expensive but increases the number of ions being introduced into the MS at a given moment in time, casting doubt on its feasibility.

### 5.3. Normalization Using Endogenous Molecules

The use of normalization using endogenous molecules has also been implemented. A particular paper by Wisniewski et al. (2014) [3] suggested that measuring DNA amount in lysed cells and using the abundance of histones as an internal standard can act as a “proteomic ruler” for normalization of results post-analysis in Perseus, as the amount of histone is proportional to DNA. However, this method may be skewing the data as not every peptide sequence that is assignable to the histones is unique, and alternative analysis pipelines, such as the PEAKS algorithm as used in our Facility, only use a restricted number of peptides to validate a hit. The proteomic ruler is based on the total ion count of the histones, but it has not been specified which peptides or how many are necessary. The PEAKS program, and likely other pipelines, will not provide the option to use any of the histones that are not significantly matched with high confidence, meaning that the user is not able to replicate the exact proteomic ruler discussed in the paper in alternative pipelines.

A variation on the above idea is normalizing post-acquisition using the ion intensity of peptides from a proteoform that is not suspected to change in abundance due to the treatment. An example of this is Protein/nucleic acid deglycase DJ-1 or ‘PARK7’ which is reported not to vary in abundance in the six mouse tissues tested or the eight human cell lines that are referenced [73]. However, other literature reports this protein to be upregulated in response to oxidative stress in renal cells [74], which were not one of the human cell lines tested. This observation should not change the fact that this is a very promising normalization technique and shows the need to survey as wide a range of proteoforms as possible. This could be achieved through data mining of repositories such as ProteomicsDB and neXtProt [75,76].

In our Core Facility, we have considered the idea of the “proteomic ruler” to raise the question as to how much the abundance of the supposedly consistent proteins like histones varies between different cell types or cell lines, and what benchmark is most suitable per cell type, especially in non-mammalian organisms such as bacteria. Particularly, we ask how much of this variation is acceptable before this means of normalization is no longer sufficient. For example, many cancers have variable numbers of chromosomes due to their genetic instability. This will affect the abundance of histones and will introduce bias between samples (i.e., between different cancers, or when comparing one type of cancer cells to healthy cells). This relates back to the idea that not all methods are appropriate for all samples [29].

## 6. Post-Analysis Normalization

When applying normalization to samples post instrument acquisition, several potential factors can affect both the consistency and quality of the generated data. Generally speaking, data level normalization is employed in order to remove inter- and intra-run variation that can be introduced as a result of the general operation of the instrument. This minor stochastic variability is expressed as “batch” effects between samples run multiple times at different times of different days then those samples will all have [77]. Furthermore, the samples would cluster by the day they were run, as has been shown via performing a principal component analysis [78]. This is in contrast to the oft-quoted idea that the same sample, even when run multiple times, will significantly cluster together [79,80].

### 6.1. Normalization of Retention Time in LC-MS

One of the first considerations that can arise when assessing reproducibility, a direct effector of normalization, is when peptides or proteoforms actually elute from the column. Minor variations in retention time can have dramatic implications when comparing the data from one run to another [78], especially when the experiment is based around Data Independent Analysis (DIA)-based Label-Free Quantification (LFQ) [81]. When performing LFQ, quantification is achieved by measuring the total area under the curve of the fragments of each detected peptide or all of the charge states of a proteoform. This means that as peptides or proteoforms are eluted off the column at the same time point, they all contribute to the total ion count and hence contribute to the absolute quantity of that peptide or proteoform [63]. The identification of the peptides themselves is inferred from a spectral library that draws its identification from a complex calculation of co-occurring peptide fragments and retention times [71], whereas proteoform identification is achieved through fragment matching [82] and accurate calculation of proteoform mass [83]. Therefore, when performing an experiment that may span several changed/replaced columns, many samples and possible buffer changes, retention time is shifted. If only the raw retention time were used then it becomes clear that the number of peptides that would provide an exact match to the library would be relatively low. When coupled with the possibility of false positives due to random matching then it is plain to see why normalization is needed.

Synthetic reference peptides have been used with some success as a means of post-acquisition normalization [71]. The general process is outlined in Figure 1. These iRT peptides elute at known and specific intervals and with known and specific intensities, and therefore can be used to align samples that would otherwise differ. These ISTDs can also be used to align datasets acquired on different generations of instruments as well as datasets that have been acquired using different chromatography gradients [84]. Though the ability to align data in this way is dependent on the previous experiment containing these iRT peptides, the benefit of having this alignment means that new and old data can be integrated together without the need to repeat entire experiments. Once retention time normalization is performed then a further layer of normalization using these iRT peptides could be performed in order to effectively “scale” the raw total area quantification data of each peptide, given knowledge of how the entire experiment, include sample preparation, was performed.

### 6.2. Approaches of Normalization of MS-Derived Data

Normalization of MS data can be approached in a number of ways depending on the experiment being performed and the overall aim of the analysis [81,85,86,87,88,89]. Experiments that focus on gaining as much coverage of the proteome as possible can benefit from methods that use “cross run” methods that seek to infer the validity of detected peptides by assessing the abundance of those peptides across an entire experiment [90]. Essentially each run is inspected for the presence or absence of peptides that are used for imputing protein identifications [89]. As each run is cross-referenced to each other run, peptides that would normally not be included in the final results can be verified by their presence in other samples. Put another way, a peptide that did not produce an MSMS spectrum can still be quantified if the MS1 mass and retention time match the same peptide in another run that has been identified by MSMS [89]. The issue with this style of normalization is that the rate of false positive identifications can rise and necessitate more extensive manual curation. Additionally, certain algorithms can decide that a particular peptide is only “real” if it is present in a minimum number of samples or in more than two of three technical replicates [91]. Though this can be a robust method for automatically curating data, there is an inherent risk for loss of biologically relevant information due to some protein identifications not meeting this stringent requirement and not being included in the final analysis. For many workflows this will not be an issue. For instance, where samples are precious or in low abundance (such as micro-dissected human-derived tissue sections), there may simply not be enough material to use high stringency identification conditions reliably [85]. As discussed earlier, the overriding assumption is that the number of peptides injected into the mass spectrometer is also normalized, as are the sample processing conditions.

When performing an LFQ analysis, normalization of area and intensity of peaks is generally applied to account for cross run variation. This is to ensure that when downstream statistical tests are performed, the variation in the area of a given peptide or proteoform charge state across multiple samples is reflective of biology and not due to simple instrumentation variation. One robust approach that can be employed is to take the median intensity of each ion across all runs and then calculate the factor that each other ion needs be multiplied by [81]. That figure can then be used to effectively “scale” the intensities of the ions against each other. The advantage to such a system is that a sample that otherwise has an overall lower ion intensity to that of its fellows can still be included in statistical tests without being labelled as an outlier. Once normalized the relative ratio between ions in that individual run can be scaled so that the metric of total quantification (in this case, total area) of each ion will be similar to that of the other samples.

### 6.3. Limitations of Normalization: When an Outlier Stays an Outlier

The prevalence of normalization techniques in mass spectrometry appears to be a natural co-evolution of high throughput “big data” experiments and the need to ensure that data from lengthy experiments are kept consistent. However, there is also the issue of deciding precisely when a particular run/sample/batch is too distinct to be able to be normalized, and the strategies for this task seem to be a matter of intuition, experience, and judgement; it is a clear that a rigorous standard is needed. When a pool of data is normalized, a measure of deviation from the mean can be calculated. Therefore, it seems logical that if a particular sample contains an error of more than 1 to 2 standard deviations of the median, the sample should be removed from the cohort, and the analysis re-performed. There are also a host of algorithmic techniques that have also been shown to be effective [85,86,90,92] however a full exploration of the operation of the mathematics behind these algorithms is beyond the scope of this review. What can be concluded is that the large quantity and diversity of computational tools suggests that no single algorithm is effective for all experiment types.

### 6.4. Cross Run Normalization, Quality Control and the Removal of “Batch Effects”

Batch effects as a phenomenon are often observed when running large experiments that span a significant period. Essentially, once a single “batch” of samples has been run, those samples will share common data features that are intrinsic to that batch. These effects result from major changes, such as in chromatography solvents, columns, IPG strip or gel casting. However, they may also be more subtle and be the result of changes to the microenvironment, such as temperature and humidity effects on the electronics of the mass spectrometer. It becomes an impossibility to ensure that the conditions on a given day are perfectly reproduced with 0% error. Instead, the resulting data generated is unique to that day. This can be a problem when LC-MS/MS acquisitions can take 3 to 4 h per injection and cannot be performed in parallel. 2D-PAGE can be performed in parallel but the greater manual handling necessary could introduce error. One approach to eliminating batch effects is to effectively normalize across all runs in an experiment. As mentioned above, the addition of spike in standards can be employed and used as a reference for all samples, however, cross run normalization can be performed by simple total protein quantity [73,93]. This works by, instead of considering the number and nature of peptides that have been detected, a simple metric of the topmost abundant proteins is used to generate a normalization factor for all proteins in a sample.

When this approach is taken or there is a high degree of expected variation (such as in the case of human tissue analysis), a quality control sample is used as a reference for the other samples in a run. Since the quality control sample is the same sample injected with each run, the relative abundance of the detected proteins should not change and hence total protein can be normalized with minimal interference [63].

Another similar solution used in labelling workflows, such as iTRAQ, TMT and DiGE, is the use of a pooled standard. This involves taking a known amount of either total protein or peptide and mixing them together in equal ratio before labelling with a single tag. This reference standard, or “master pool” [94] is then added to the other individually labelled samples and used to normalize that other samples. It is most often used to enable the comparison of different iTRAQ/TMT experiments where, for example, fourteen samples are quantified in two iTRAQ 8-plex experiments where seven channels in one experiment are individual samples and the last channel is the pooled standard of the fourteen samples. Intuitively, this approach should provide robust normalization. However, it also has the potential to suffer bias from the sources already listed in this review, such as errors in protein assay. The master pool approach has also been used for the creation of spectral libraries for DIA acquisition to reduce the time required for library construction but has not otherwise been applied to MS-based LFQ.

## 7. Conclusions

At this point, we return to the original question: When in the experimental procedure should normalization be applied? Ensuring that the number of cells in each sample to be extracted, be it control or treatment, is identical is one way of normalizing but do changes in cell size and volume need to be considered? A larger cell volume would imply a greater number of proteins and thus a population of cells that has an altered cell volume after treatment would not be able to be normalized by cell number. A related question to this is how much accuracy is required in the counting of cell number to ensure adequate normalization, which is related to the detection limit of the technique in question. Put more simply, how many cells are needed to yield enough of a proteoform or peptide to be confidently detected. Again, this is going to vary with the complexity of the sample, and less fractionated samples will require a proteoform or peptide to be in a higher concentration for more confident detection. Protein copy per cell, or more accurately ORF product copy per cell, is known in some systems, providing this method with strong potential [4,95].

The most prevalent way of normalization seems to be by total protein amount as estimated by a protein assay and this is widely applied in Top Down and Bottom Up approaches. This relies on the majority of the proteome not being altered in abundance due to the treatment being studied or on the concentration of highly abundant proteoforms not changing, something that cannot be assessed until after the experiment has been performed. An example of this from our Core Facility is Seagrass [96], which graphically demonstrates how extreme differences in the abundance of particular proteoforms can be when treated just with higher intensity light. The gels are loaded with the same total amount of protein but the software is unable to quantify any abundance changes because there are too few spots displaying the same abundance to which normalization can be performed. The solution that presents itself is to perform normalization to a proteoform whose abundance does not change under any circumstances, an idea presented earlier in this article. There are a number of potential candidate proteins but normalization is being applied post-acquisition to a sample previously normalized by total protein load. Whether more bias is being introduced by post-acquisition normalization of a sample that is potentially already biased has not been tested and should be investigated. It should be assessed whether normalization by predetermination of the abundance of ‘reference’ proteoforms that do not alter in abundance is possible and useful. This would require pre-analysis of the sample to determine the normalization to be applied which lengthens the experimental time and reagents required.

## Figures and Tables

**Figure 1 proteomes-07-00029-f001:**
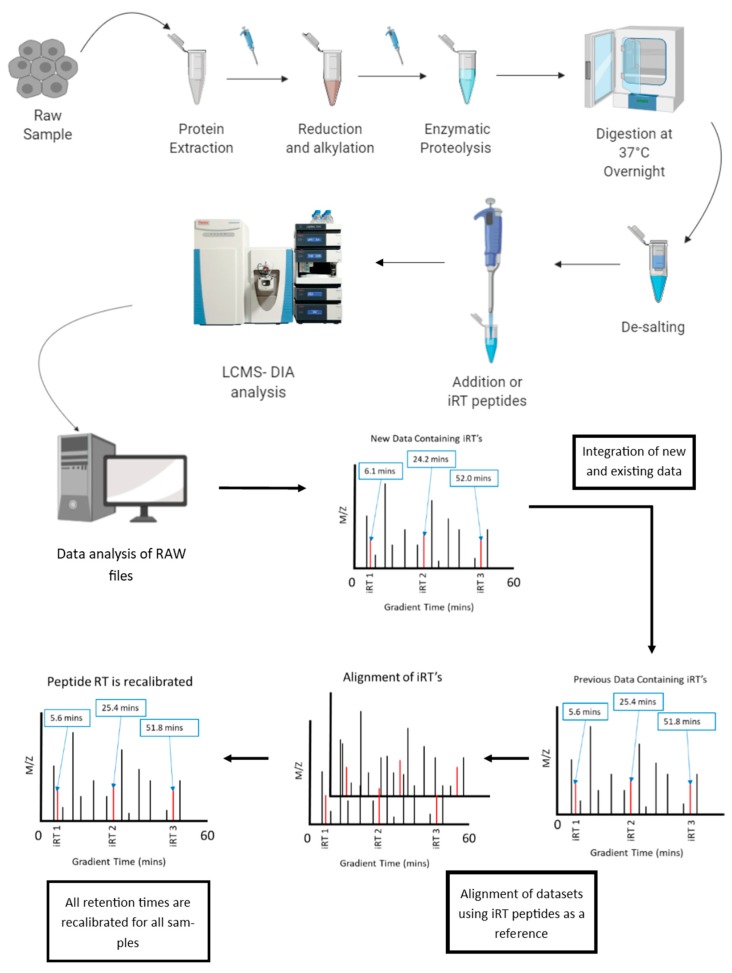
Normalization of Retention Time in LC-MS/MS using Synthetic Reference Peptides, like Indexed Real Time (iRT) peptides. After protein extraction and digestion of the raw sample into peptides, peptides are desalted and iRT peptides are added just prior to LCMS-DIA analysis as a means of normalizing by retention time.

**Table 1 proteomes-07-00029-t001:** Comparison of the Common Protein Quantification Methods by analysis of positive and negative aspects.

Category	Method	Positives	Negatives
Fluorometric	UV absorption (Tryptophan)	Rapid Low cost Sample recoverable for proteomic preparations	Quantitates only the amino acids tyrosine, tryptophan and phenylalanine Sensitive to detergents
	Qubit Protein Assay	Sensitive at low protein concentrations Small sample volumes (≤10 µL)	Sensitive to temperature fluctuations Easily saturated
Colorimetric	BCA	Compatible with detergents at low concentrations	Quantitates only the amino acids tyrosine, tryptophan and cysteine Sample not recoverable Sensitive to detergents
	Bradford/Coomassie	Compatible with reducing agents Reagent binds to protein rather than to individual amino acids	Sample not recoverable
	Lowry	Sensitive	Timely and laborious procedure Sample not recoverable Sensitive to detergents and other common reagents
Densitometry	SDS-Page (In-gel)	Highly accurate Sample recoverable for proteomic preparations however, laborious process	Analysis susceptible to bias depending on gating of bands
	Western-Blot and ELISA (also considered Flourometric or Colorimetric depending on tag or application)	Target-protein specific	Analysis susceptible to bias depending on gating of bands

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
