# Peer review of "What is Normalization? The Strategies Employed in Top-Down and Bottom-Up Proteome Analysis Workflows"

_proteomes, 2019, doi:10.3390/proteomes7030029_

Round 1
Reviewer 1 Report
I found the review to be well written and useful. I think it would have been helpful to have a few specific examples of actual clinical tests where the methods you describe are used, especially the gel electrophoretic approaches. Second, I think something that was not mentioned but that is very important is how to prevent chemical alterations to analytes during the collection and processing period, e.g. proteolytic enzyme inhibitors for protein or peptide analysis and potentially anti-oxidants for lipid analysis. At a minimum they should be mentioned and useful references provided.
Author Response
Reviewer 1:
I found the review to be well written and useful.
I think it would have been helpful to have a few specific examples of actual clinical tests where the methods you describe are used, especially the gel electrophoretic approaches.
There are many proteomic techniques that are not directly applied clinically, but are used to find potential ‘biomarkers’ which are then utilised in more easily applied techniques such as ELISAs. Thus, there are no specific examples of current clinical tests as electrophoretic approaches were replaced long ago.
Second, I think something that was not mentioned but that is very important is how to prevent chemical alterations to analytes during the collection and processing period, e.g. proteolytic enzyme inhibitors for protein or peptide analysis and potentially anti-oxidants for lipid analysis. At a minimum they should be mentioned and useful references provided.
Response: This idea is alluded to in part 3 lines 162-166 but there are no specific examples. We have now added a section from 177-182 that address this comment. We have also referred to the example of carbamylation as an example of an avoidable chemical modification. We have not included an example for lipids, such as the inclusion of butylated hydroxytoluene as an anti-oxidant because lipids are not the focus of the manuscripts. We acknowledge that many of the points we make in this manuscript equally apply to metabolomics and lipidomics.
Reviewer 2 Report
This is a review of methods and issues related to the problem of normalization in proteomics techniques.
1) The authors define their paper as:” a starting point for discussion of the definition of and the issues surrounding the concept of normalisation” (abstract, lines 23-24). This statement is too generic: normalization is a generic term applied in different fields (from sociology to quantum mechanics) with different meanings. Limiting to statistics and its application, normalization can have a range of different meanings. Indeed, the tendency of authors to introduce the term in a very general way is also reflected by the definition provided at line 38: “normalization is defined as the process of returning something to a normal condition or state”. A more pertinent definition is reported in the next three lines:” In system-wide -omics analysis, normalisation tries to account for bias or errors, systematic or otherwise, to make samples more comparable [1] and measure differences in the abundance of molecules that are due to biological changes rather than bias or errors”. It is clear from this sentence the authors mean to discuss normalization methods aimed to correct “bias or errors, systematic or otherwise”. However, in most text “bias” is defined as a systematic error and normalization methods are aimed to account for bias. In conclusion this referee thinks that the definition of normalization should be limited to the specific purpose (methods to correct systematic errors (or bias) in proteomic analyses) and introduction should be rewritten.
2) Moreover, in several sections (i.e. 4.2. Digestion optimisation to reduce bias, 4.3. Monitoring digestion efficiency) the authors seem to discuss experimental details that avoid introducing bias during data acquisition, more than normalization procedure that try to correct a systematic bias in the data. In general, the authors should distinguish between experimental strategies or workflows aimed to avoid the introduction of specific systematic errors and the mathematical methodology (for instance median scaling, central tendency, linear regression, locally weighted regression, and quantile techniques) or experimental strategy (SILAC) aimed to correct systematic errors in the post-acquisition phase.
3) Mathematical methodology is poorly discussed. For instance, the basic concept of proteomic ruler is not explained in mathematical terms. Several mathematical techniques for normalization are not covered in the text.
4) In Section 7.3. (Limitations of normalisation: when an outlier stays an outlier) the authors discuss about “outlier removal” that is not part of normalization techniques according to the strict definition (techniques dedicated to correct systematic errors).
Minor points:
Proteomic ruler (Wisniewski et al 2014) is erroneously indicated as proteome ruler.
Some abbreviations are introduced in the text before of their explanation (i.e. LFQ).
Figures and schematic illustrations are needed to clarify some points.
Description at line 429-436 should be rewritten: a figure can be useful.
Author Response
This is a review of methods and issues related to the problem of normalization in proteomics techniques.
1 - The authors define their paper as:” a starting point for discussion of the definition of and the issues surrounding the concept of normalisation” (abstract, lines 23-24). This statement is too generic: normalization is a generic term applied in different fields (from sociology to quantum mechanics) with different meanings. Limiting to statistics and its application, normalization can have a range of different meanings. Indeed, the tendency of authors to introduce the term in a very general way is also reflected by the definition provided at line 38: “normalization is defined as the process of returning something to a normal condition or state”.
A more pertinent definition is reported in the next three lines:” In system-wide -omics analysis, normalisation tries to account for bias or errors, systematic or otherwise, to make samples more comparable [1] and measure differences in the abundance of molecules that are due to biological changes rather than bias or errors”. It is clear from this sentence the authors mean to discuss normalization methods aimed to correct “bias or errors, systematic or otherwise”. However, in most text “bias” is defined as a systematic error and normalization methods are aimed to account for bias. In conclusion this referee thinks that the definition of normalization should be limited to the specific purpose (methods to correct systematic errors (or bias) in proteomic analyses) and introduction should be rewritten.
Response: This comment is well received, one of our key points was that the definition of normalisation is hard and broad. We do acknowledge that by describing this we have actually failed to narrow down a workable definition for the purpose of this review. Therefore, we have added a few lines to the abstract and readjusted the introduction (Lines 52-64 and 71). One point mentioned that we disagree with, is to limit the use of the term normalisation to just the correction of bias or systematic errors. In our opinion, normalisation is less a strict term and more of an umbrella term used to describe a number of techniques and approaches used to achieve a central goal which is to try and ensure that bias from any source is eliminated. We also acknowledge that this may differ in fields outside of proteomics, therefore we have limited the application of our definition of normalisation to proteomics and other -omics style analyses.
2 - Moreover, in several sections (i.e. 4.2. Digestion optimisation to reduce bias, 4.3. Monitoring digestion efficiency) the authors seem to discuss experimental details that avoid introducing bias during data acquisition, more than normalization procedure that try to correct a systematic bias in the data.
Response: This point should now be addressed with the amendments mentioned above.
In general, the authors should distinguish between experimental strategies or workflows aimed to avoid the introduction of specific systematic errors and the mathematical methodology (for instance median scaling, central tendency, linear regression, locally weighted regression, and quantile techniques) or experimental strategy (SILAC) aimed to correct systematic errors in the post-acquisition phase.
Response: Again, this should be now be rectified.
3 - Mathematical methodology is poorly discussed. For instance, the basic concept of proteomic ruler is not explained in mathematical terms. Several mathematical techniques for normalization are not covered in the text.
Response: Our answer to this point is simply that our review does not extend to the various mathematical or algorithmic methods that are available through various software packages or R scripts. Our aim for this review was to provide a well explained and comprehensive guide to the broad range of normalisation strategies that exist. The in depth specifics of the mathematics used for various strategies are not within the scope or intention of this manuscript. Those are well described in numerous scientific articles and reviews.
In Section 7.3. (Limitations of normalisation: when an outlier stays an outlier) the authors discuss about “outlier removal” that is not part of normalization techniques according to the strict definition (techniques dedicated to correct systematic errors).
Response: The definition of normalisation has been adjusted (as mentioned previously) so this is now encompassed.
Minor points:
Proteomic ruler (Wisniewski et al 2014) is erroneously indicated as proteome ruler.
Response: This has been amended throughout the manuscript.
Some abbreviations are introduced in the text before of their explanation (i.e. LFQ).
Response: We believe the reviewer was mistaken in this regard, The example given appears at line 410 and is immediately preceded by the wording
“…experiment is based around data independent analysis (DIA)-based label-free quantification (LFQ).
This is the first use of LFQ and is the sentence describing it.
Figures and schematic illustrations are needed to clarify some points.
Response: The reviewer has not suggested what points would be clarified by including figures. We attempted to find places in the manuscript where figures could help but decided that they did not add any clarification to any points and would, in some cases, complicate points.
Description at line 429-436 should be rewritten: a figure can be useful.
Response: A figure has been added at section 7.1.
Reviewer 3 Report
This review is very interesting since the topic is very current in proteomics.
It provides a general overview of the techniques/strategies adopted to perform protein normalization in a proteomic workflow. I like very much the idea of this review as a starting point for a more enlarged discussion about the concept of normalization.
As regards paragraph 4.1. Normalization using total protein quantification, I would suggest the author to consider also the possibility to normalize total amount of protein by means of 1D SDS PAGE analyses employing protein Coomassie blue staining or other quantitative methods in combination with the use of calibration curves. These methods have the drawback of being time consuming but they circumvent the interference of some substances that are usually encountered with in-solution quantification methods.
My only additional suggestion to the authors is to do an additional effort in providing through the review some schemes/figures in order to summarize important points. Indeed, sometimes a figure provides a clearer and more direct message with respect than a long paragraph.
Author Response
This review is very interesting since the topic is very current in proteomics.
It provides a general overview of the techniques/strategies adopted to perform protein normalization in a proteomic workflow. I like very much the idea of this review as a starting point for a more enlarged discussion about the concept of normalization.
As regards paragraph 4.1. Normalization using total protein quantification, I would suggest the author to consider also the possibility to normalize total amount of protein by means of 1D SDS PAGE analyses employing protein Coomassie blue staining or other quantitative methods in combination with the use of calibration curves. These methods have the drawback of being time consuming but they circumvent the interference of some substances that are usually encountered with in-solution quantification methods.
Response: We did make a mention of Coomassie blue staining however that was not property elaborated on. We have added a section to expand this. It was remiss of us not to include greater emphasis on the most commonly utilised assay in our facility.
My only additional suggestion to the authors is to do an additional effort in providing through the review some schemes/figures in order to summarize important points. Indeed, sometimes a figure provides a clearer and more direct message with respect than a long paragraph.
Response:This has been addressed in the response for reviewer 2.
Reviewer 4 Report
In this manuscript, the authors pointed out various factors hampering accuracy of quantitative performance in proteomic experiments. Researchers in this field often face these problems, but such problems are not often discussed in review articles. This manuscript is valuable in that it listed a variety of points to consider in a quantitative analysis. On the other hand, as mentioned in the introduction, it does not provide a general solution to overcome these problems because experimental designs are so different from one experiment to another. My personal opinion is that, in most case, total protein amount would be a good value for normalization, although there should numerous problems. In addition, this manuscript would be better if it could be more concise by omitting small points. Specific points are as follows.
Chapter 1
1) 59. What is "Cellular normalisation" ? Does this include normalization methods based on protein concentration, cell number and DNA amount ?
2) It is to be clarified how SILAC overcomes the problems in "Cellular normalization". Doesn't SILAC needs normalization by protein amount, cell number or else ? If so, SILAC is similar to, but not superior to other methods in terms of normalization, isn't it?. In addition, SILAC has a drawback in that it can only be applied to cultured cells in a real situation.
Chapter 5
3) 259. Quantitation by isobaric tag labelling is not necessary "absolutely" rely on protein determination, because small differences in total protein amounts could be normalized by the median, average or total sum of ion counts of each samples. Actually, using softwares such as MASCOT, we can choose one of these normalization methods.
Author Response
In this manuscript, the authors pointed out various factors hampering accuracy of quantitative performance in proteomic experiments. Researchers in this field often face these problems, but such problems are not often discussed in review articles. This manuscript is valuable in that it listed a variety of points to consider in a quantitative analysis. On the other hand, as mentioned in the introduction, it does not provide a general solution to overcome these problems because experimental designs are so different from one experiment to another. My personal opinion is that, in most case, total protein amount would be a good value for normalization, although there should numerous problems.
Response: We welcome the opinion of the reviewer. Our reasoning for writing this manuscript is to start some discussion and the reviewer’s comments would indicate our success.
In addition, this manuscript would be better if it could be more concise by omitting small points. Specific points are as follows.
Chapter 1
1) 59. What is "Cellular normalisation" ? Does this include normalization methods based on protein concentration, cell number and DNA amount ?
Response: This is explained in the section entitled ‘cellular normalisation’.
2) It is to be clarified how SILAC overcomes the problems in "Cellular normalization". Doesn't SILAC needs normalization by protein amount, cell number or else ? If so, SILAC is similar to, but not superior to other methods in terms of normalization, isn't it?. In addition, SILAC has a drawback in that it can only be applied to cultured cells in a real situation.
Response: The reviewer is correct in his comments and we have added a line pointing out that the overall considerations in the manuscript also apply to SILAC.
3) 259. Quantitation by isobaric tag labelling is not necessary "absolutely" rely on protein determination, because small differences in total protein amounts could be normalized by the median, average or total sum of ion counts of each samples. Actually, using softwares such as MASCOT, we can choose one of these normalization methods.
Response: While the reviewer is correct and small differences can be dealt with post-acquisition, our argument is that normalisation and the removal of bias should be considered at the very beginning so that post-acquisition normalisation can actually perform better.
We respect the opinion of reviewer 4 however our aim was to explain and discuss a broad range of techniques rather than picking a single one as the “best” approach. Additionally, we have discussed many of the reviewer’s points with both the positive and negative aspects of the techniques mentioned. Again, we were not aiming to choose the best approach. Our genuine opinion is that no individual normalisation method is a fix-all for all workflows and we believe that we have effectively communicated this point.
Round 2
Reviewer 2 Report
The authors addressed adequately all the issues